# Simultaneous leaf-level measurement of trace gas emissions and photosynthesis with a portable photosynthesis system

Mj Riches, Daniel Lee, Delphine K. Farmer

Department of Chemistry, Colorado State University, Fort Collins, 80523, USA

*Correspondence to*: Delphine K. Farmer (Delphine.Farmer@colostate.edu)

**Abstract.** Plants emit considerable quantities of volatile organic compounds (VOCs), the identity and amount of which vary with temperature, light and other environmental factors. Portable photosynthesis systems are a useful method for simultaneously quantifying in situ leaf-level emissions of VOCs and plant physiology. We present a comprehensive characterization of the LI-6800 portable photosynthesis system's ability to be coupled to trace gas detectors and measure
leaf-level trace gas emissions, including limits in flow rates, environmental parameters, and VOC backgrounds. Instrument contaminants from the LI-6800 can be substantial, but are dominantly complex molecules such as siloxanes that are structurally dissimilar to biogenic VOCs and thus unlikely to interfere with most leaf-level emissions measurements. We validate the method by comparing $CO_2$ assimilation calculated internally by the portable photosynthesis system to measurements taken with an external $CO_2$ gas analyzer; these assimilation measurements agree within 1 %. We also
demonstrate both online and offline measurements of plant trace gas exchange using the LI-6800. Offline measurements by pre-concentration on adsorbent cartridges enable detection of a broad suite of VOCs, including monoterpenes (e.g., limonene) and aldehydes (e.g., decanal). Online measurements can be more challenging if flow rates require dilution with ultra-pure zero air. We use high resolution time-of-flight chemical ionization mass spectrometry coupled to the LI-6800 to measure direct plant emission of formic acid.

## 1 Introduction

Non-methane volatile organic compounds (VOCs) are readily oxidized in the atmosphere and thus impact atmospheric composition, climate and human health. As such, a quantitative understanding of VOC sources is essential for predicting future air quality and climate conditions. VOC oxidation impacts greenhouse gas mixing ratios by both producing tropospheric ozone and lowering OH radical mixing ratios, thereby increasing the lifetime of atmospheric methane
(Kesselmeier and Staudt, 1999). Oxidized products of VOC precursors contribute to secondary organic aerosol (Faiola et al., 2018), which impacts climate and human health (Davidson et al., 2005; Pope III and Dockery, 2006). Biogenic emissions from plants dominate the global VOC source (Guenther et al., 1995; Lamarque et al., 2010; Lathière et al., 2006); terrestrial ecosystems and the ocean emit 1150 TgC yr$^{-1}$ of VOCs globally (Guenther et al., 1995), relative to anthropogenic VOC sources, which account for only 142 TgC yr$^{-1}$ globally (Singh, 1995). The most abundant group of biogenic VOCs (hereafter

"BVOCs") are isoprenoids (Kesselmeier and Staudt, 1999), molecules comprised of $(C_5H_8)_n$ units. Isoprene $(C_5H_8)$ contributes to roughly half of global BVOC emissions, while monoterpenes $(C_{10}H_{16})$ and sesquiterpenes $(C_{15}H_{24})$ account for an additional 18% combined (Guenther et al., 2012).

BVOC emissions are affected by a complex combination of factors, including temperature (Tingey et al., 1980; Duhl et al.,
2008; Tarvainen et al., 2005; Sharkey and Yeh, 2001), soil moisture (Ebel et al., 1995; Ormeño et al., 2007; Sharkey and Loreto, 1993), light (Tarvainen et al., 2005; Sharkey and Loreto, 1993; Owen et al., 2002; Staudt and Seufert, 1995), $CO_2$ concentration (Wilkinson et al., 2009; Loreto and Schnitzler, 2010), plant developmental stage (Holopainen, 2004; Kim et al., 2005; Zhang and Chen, 2009; Guenther, 1997), mechanical stress (Kaser et al., 2013a; Markovic et al., 2016), and biotic stress (Mauck et al., 2010; Niinemets et al., 2013; Scala et al., 2013). While the effects of some environmental factors, such
as temperature, are well-understood, the effects of other factors, such as $CO_2$ concentration, are less clear. Different VOCs also have different temperature responses, and different plant species have different temperature responses for the same VOC. While most VOC emissions increase exponentially with a linear increase in temperature (Tingey et al., 1990; Peñuelas and Llusià, 2001; Niinemets et al., 2004) before reaching a maximum and rapidly decreasing (Grote et al., 2013), others are not sensitive with temperature (e.g., cis-β-ocimene) (Loreto et al., 1998). Temperature effects on VOC emissions are
included in emission models, typically based on the results of short-term exposure experiments (Guenther et al., 1993; Guenther et al., 2012). Unlike temperature, the effect of changing $CO_2$ concentrations on BVOC emissions is under debate, even among plants of the same species (Loreto and Schnitzler, 2010). Under elevated $CO_2$ conditions, some studies observe no change in emissions (Constable et al., 1999; Kainulainen et al., 1998; Räisänen et al., 2008; Rapparini et al., 2003), while others observe a decrease (Scholefield et al., 2004; Sallas et al., 2003; Snow et al., 2003) or increase (Staudt et al., 2001a) in
VOC emissions relative to ambient $CO_2$. Despite its importance to atmospheric composition, biogenic VOC emission response to environmental change remains poorly understood.

Global emission inventories of BVOCs vary across models (Arneth et al., 2008; Grote et al., 2013). Monoterpenes are treated less consistently than isoprene: the standard deviation of monoterpene emissions across multiple emission models is
40% of the mean, compared to 10% for isoprene (Arneth et al., 2008). Emission models that group several VOCs together, such as the monoterpene isomers, may simplify the model, but this approach assumes that emissions are similar across the isomeric class and neglects differences in atmospheric reactivities of compounds. For example, the lifetime for reaction with ozone between α-pinene and β-pinene differ between a few hours to a day (Atkinson and Arey, 2003), which consequently affects the SOA yield (Friedman and Farmer, 2018). Some models use plant photosynthesis to predict VOC emissions (Grote
et al., 2013; Grote et al., 2014), though the correlation between plant physiology and VOC emission – let alone the response of these parameters to external environmental stressors – is not well understood. Model limitations are due, in part, to the limited availability of measurements, particularly simultaneous measurements of plant physiology and speciated VOC emissions.

VOC emissions are commonly quantified through canopy measurements (e.g., Goldstein et al., 2004; Rinne et al., 2007; Kaser et al., 2013b; Ciccioli et al., 1999) and leaf or branch chamber headspace measurements (e.g., Kessler and Baldwin, 2001; Llusià et al., 2002; Komenda et al., 2001; Guenther et al., 2000). One approach to leaf-level studies couples a portable photosynthesis system (PPS) with a trace-gas analyzer, thus enabling simultaneous physiology and VOC emissions measurements (e.g. Lerdau and Keller, 1997; Brilli et al., 2007; Singsaas et al., 1999; Loreto and Velikova, 2001; Geron et

al., 2006b; Brilli et al., 2011; Harley et al., 2014). The user can clamp the cuvette of the PPS onto a leaf and thereby control leaf-level parameters such as light wavelength and intensity, leaf temperature, humidity, air flow, and $CO_2$. Within the PPS, two infrared gas analyzers (IRGAs) determine the difference in gas concentration of $CO_2$ and water before and after the leaf cuvette. The system calculates physiological parameters including $CO_2$ assimilation rate (A), transpiration and stomatal conductance (for detailed calculations, refer to LI-COR, 2017). The $CO_2$ assimilation rate refers to the rate of photosynthetic

$CO_2$ uptake into the leaf, transpiration is the rate at which water vapor is released from a leaf, and stomatal conductance is the rate at which $CO_2$ and water pass through the stomata of a leaf. Diverting the PPS air flow to an external gas analyzer enables users to sample leaf emissions. Emissions analysis can be both *in situ* and real-time if online detection techniques are available, such as proton transfer reaction mass spectrometry (PTR-MS; e.g. Brilli et al., 2011; Brilli et al., 2007; Harley et al., 2014) or portable gas chromatography (e.g., Geron et al., 2006b; Lerdau and Keller, 1997; Singsaas et al., 1999; Loreto

and Velikova, 2001). However, gas samples can also be collected for offline analysis by thermal desorption gas chromatography mass spectrometry (e.g., Geron et al., 2006b; Harley et al., 2014) and gas chromatography mass spectrometry canister analysis (e.g., Geron et al., 2006a). These PPS-coupled techniques allow users to simultaneously obtain plant photosynthesis metrics and leaf-level VOC emissions.

While the PPS-VOC sampling technique has been used for decades, recent developments in PPSs provide new opportunities for leaf-level BVOC emission studies. The expanded ability to control environmental parameters, including leaf vapour pressure deficit, provides ample opportunity to study the connection between plant physiology and emission. However, PPS systems have not been rigorously evaluated in the literature for leaf-level emissions. Here, we characterize the recently developed LI-6800 portable photosynthesis system for leaf-level emissions, quantifying the capabilities and limitations of

this method. We investigate the instrumental limits of this approach, including acceptable flow rates and best practices. We demonstrate the utility of this technique for offline measurements using thermal desorption gas chromatography mass spectrometry and online measurements using time-of-flight chemical ionization mass spectrometry.

## 2 Instrumentation

We use a commercial portable photosynthesis system (LI-6800) with a Multiphase Flash™ Fluorometer (LI-COR, Nebraska)

for $CO_2$ and $H_2O$ gas exchange measurements. The PPS consists of two major components: the console, which includes the

digital interface and the chemical columns for control of air composition; and the head, which contains the 6 cm$^2$ leaf chamber and controls leaf temperature. The LI-6800 PPS controls environmental conditions at the leaf level, including: temperature, humidity, light intensity and wavelength, and $CO_2$. The PPS also controls air flow and fan speed. As described in the Introduction, the PPS uses IRGAs to detect gas concentrations of $CO_2$ and water from before (reference, REF) and after (sample, SAM) the leaf chamber. The LI-6800 PPS has ports on both of these sample lines; air collected from the REF subsampling port can be used as a system background for emissions that do not occur within the PPS itself, while air collected from the SAM port is representative of leaf emissions and the system background. In instances where the analytes of interest are only emitted by plant tissue and not by the PPS, measurements taken from the REF port can be used to subtract background from the SAM port samples.

We define our standard operating conditions in Table 1, along with the technical capabilities of the instrument and the acceptable range determined herein. We acquired response curves by altering a single environmental parameter (e.g., temperature), waiting for leaf photosynthesis (i.e. $CO_2$ assimilation) to stabilize to new conditions, and then collecting gas exchange and VOC measurements. To determine the parameters for photosynthesis stabilization, we monitored a leaf using the PPS for 20 minutes, and determined the natural variability in stomatal conductance and $CO_2$ assimilation. A standard deviation limit can be set for the stability parameters, but we found the natural variability in our citrus plants changes daily. Therefore, we determined stability using a limit on the slope of stomatal conductance (0.01 mol m$^{-2}$ s$^{-1}$ min$^{-1}$) and $CO_2$ assimilation (0.5 µmol m$^{-2}$ s$^{-1}$ min$^{-1}$) measurements over a 15 second period. Photosynthesis stabilization took anywhere from 30 seconds to 15 minutes, depending on how close the set environmental conditions were to ambient or prior conditions. Unless otherwise noted, we controlled the LI-6800 input gas stream with a $CO_2$ scrubber (soda lime, LI-COR 9964-090), dessicant (blue-indicating Drierite, LI-COR 622-04299), humidifier (Stuttgarter Masse, LI-COR 9968-165), and $CO_2$ (8 g cartridges, LI-COR 9968-227 and Leland 30404). The values for flow rate and chemical conditions are in Table 1; further details on the instrument specifications, including component precision, can be found in the instrument manual (LI-COR, 2017).

Note that the LI-6800 denotes flow in terms of µmol s$^{-1}$. All flows are given in L min$^{-1}$; we performed experiments at 1525 m above sea level and use an air pressure of 0.844 atm for conversion calculations when necessary.

The flow path of the PPS subsampling system is shown in Fig. 1. Ambient air is pulled into the PPS through the air inlet between 1.18 and 2.96 L min$^{-1}$ (680-1700 µmol s$^{-1}$), and is then treated for humidity and $CO_2$. The bulk flow is automatically calculated by the PPS software to control the user-defined parameter for chamber air flow (described in Table 1). A subsample of this ambient air flows through the REF IRGA and when in use, the REF subsampling port, while the remaining air enters the leaf chamber. Air exiting the leaf chamber is split between the SAM subsampling port and the second SAM IRGA. Air from the SAM and REF IRGAs is removed as exhaust through the main exhaust line. During emissions sampling,

the subsampling ports of the PPS can be simultaneously connected to trace gas analyzers, or alternated between a single analyzer with the other subsampling port closed. The air flow drawn out of the subsampling ports vary depending on emission sampling technique, and is described in more detail in Sect. 2.2.

The LI-6800 can be used with both online and offline emission sampling techniques. We use a chemical ionization mass
spectrometer (CIMS) and an external $CO_2$ detector for online sampling, but note that the principles of flow rate control are easily generalized for other trace gas analysis including PTR-MS. We use thermal desorption gas chromatography mass spectrometry for offline analysis. These systems are described in detail below.

## 2.1 Portable Photosynthesis System

The PPS consists of the head (i.e., the device which clamps onto a leaf) (Fig. 2) and the console (i.e., the device which
regulates environmental conditions and chemical use). The leaf chamber (Fig. 2A) was left unchanged while trace gas detector manifolds were connected to the SAM and REF subsampling ports (Fig. 2B and 2C, respectively). A 3.175 mm brass hose barb fitting is attached to each of the subsampling ports, followed by a 38 mm piece of flexible tubing (Tygon™, 6.35 mm o.d., 3.175 mm i.d.) that connects to a 1/4" stainless steel tee (Ultra-Torr). On each of the remaining ports (one perpendicular (Fig. $2B_2$, $C_2$) and one lateral (Fig. $2B_1$, $C_1$)), a 38 mm piece of polytetrafluoroethylene (PTFE) tubing (6.35
mm o.d., 3.175 mm i.d.) connects to a 6.35 mm perfluoroalkoxy alkane (PFA) fitting. The PFA fittings are capped unless actively used. For sorbent tube sampling, a cap on the lateral port (Fig. $2B_1$ for SAM, $C_1$ for REF) is replaced with a 6.35 mm fitting, and the sorbent tube (Fig. 2D) is fit directly in line. The external pump (Fig. 2E) is placed downstream of the tube and ensures constant flow through the sorbent tube.

When subsampling the PPS air for BVOC emissions, an external pump subsamples air through the REF and/or SAM subsampling ports. The external pump ensures constant flow through the BVOC measurement system. The bulk flow through the system ($F_I$) is controlled by an internal pump in the console and any additional pumps used by trace gas analyzers on the REF or SAM subsampling ports. Thus the total air inlet flow is the sum of flows through REF port ($F_R$), SAM port ($F_S$) and the exhaust ($F_E$):

$F_I = F_R + F_S + F_E$                                                                   (1)

$F_E$ includes flow from the internal REF and SAM IRGAs. The IRGAs each require at least 0.17 L min$^{-1}$ (100 µmol s$^{-1}$) - though a flow above 0.35 L min$^{-1}$ (200 µmol s$^{-1}$) is preferential - and the inlet flow can be a maximum of 2.96 L min$^{-1}$ (1700 µmol s$^{-1}$). Due to the instrumental limitations of these flows, sampling flows ($F_R$ and $F_S$) must not reach so high as to interfere with PPS system function. For thermal desorption sampling, where flow rates typically reach 0.2 L min$^{-1}$, samples
can simultaneously be collected through both subsampling ports. The instrument will automatically calculate the split of flows between the IRGAs to account for system requirements. While higher flows (e.g., 1 L min$^{-1}$) can be sampled via the subsampling ports, the user will need to manually adjust the flow splits using the digital user interface on the console (LI-

COR, 2017). Using higher flow rates to accommodate sampling from the SAM port will impact the flow through the leaf chamber, and thus the conditions experienced by the leaf tissue. Impact of increased flow rates should be investigated for individual species.

## 2.2 Online measurements : TOF-CIMS

The PPS trace gas sampling scheme described above is well-suited for online trace gas detection. Here, we use two systems: (1) a CO2 analyzer and (2) a high resolution time-of-flight chemical ionization mass spectrometer (TOF-CIMS; Aerodyne Research Inc. and Tofwerk AG) (Brophy and Farmer, 2015) coupled to iodide reagent ions (Lee et al., 2014) to detect gas-phase formic acid. Details of the TOF-CIMS are in S1.

For external comparison of leaf $CO_2$ exchange with the internal IRGAs, we use an external $CO_2$ analyzer (LI-840A, Li-Cor, Nebraska), which was alternately connected to the REF and the SAM subsampling ports. The LI-840A analyzer requires 1L $min^{-1}$ of flow.

The TOF-CIMS pulls 1.9 L $min^{-1}$, exceeding the maximum threshold for the PPS subsampling ports. To decrease the flow, we dilute the subsampled air with $2.00 \pm 0.05$ L $min^{-2}$ of ultra-high purity zero air (UZA; Airgas) at the inlet to the CIMS. The diluting flow is controlled by a mass flow controller (MKS Instruments, Mass Flo® Controller, 1179B).

We calculate formic acid emission rates as follows:

$$C_P = C_C * \frac{Q_C}{Q_P} \tag{2}$$

where $C_P$ is the mixing ratio of the VOC coming from the PPS (mol $mol^{-1}$), $C_C$ is the mixing ratio of the VOC identified by the CIMS (mol $mol^{-1}$), $Q_C$ is the total flow pulled by the CIMS (L $min^{-1}$), and $Q_P$ is the flow taken from the PPS subsampling port (L $min^{-1}$). To get $C_C$, a calibration is used to convert integrated peak area into concentration; the resulting value is then divided by the time over which the integration occurred.

We then convert the leaf chamber flow ($Q_L$) from L $min^{-1}$ to mol $min^{-1}$ using:

$$Q_L(mol\ min^{-1}) = \frac{Q_L(L\ min^{-1})*P}{R*T} \tag{3}$$

where P is atmospheric pressure, R is the gas constant, and T is air temperature. Using equations 2 and 3, we obtain:

$$E_{VOC} = \frac{C_P*Q_L}{S} \tag{4}$$

where $E_{VOC}$ is the VOC emission rate (mol $m^{-2}$ $min^{-1}$), and S is the leaf area ($m^2$).

**2.3 Offline detection: sorbent tubes**

Thermal desorption (TD) gas chromatography mass spectrometry (GC/MS) is an offline sampling technique commonly used to sample atmospheric volatile and semi-volatile organic compounds (Harper, 2000). This technique pre-concentrates trace gases on sorbent tubes, which are stainless steel or glass tubes of specific dimensions that are filled with adsorbent materials. Different adsorbents target different analytes. Tenax TA is a general adsorbent, which has a sampling range of 7 to 26 carbons ($C_7$-$C_{26}$), and is relatively hydrophobic (Dettmer and Engewald, 2002). Other adsorbents, such as carbon molecular sieves (e.g., Carboxen 563) collect smaller molecules ($C_2$-$C_5$), but are sensitive to atmospheric humidity (Dettmer and Engewald, 2002). As air flows through the sorbent tubes, atmospheric constituents adsorb onto the surface. The tubes are then rapidly heated and the compounds thermally desorbed into an air stream for analysis by GC/MS. Here we use an autosampler (Ultra-xr, Markes Intl.) and thermal desorption unit (Unity-xr, Markes Intl.) coupled to a gas chromatograph (TRACE 1310, Thermo Scientific) mass spectrometer (TSQ 8000 Evo Triple Quadrupole GC-MS/MS, Thermo Scientific).

Details of the TD-GC/MS method are in S2. Briefly, we use the TD-GC/MS with Tenax adsorbent cartridges to quantify seven monoterpenes, summarized in Table 2.

We calculate leaf-level VOC emissions from the cartridge samples as follows:

$$E_{VOC} = \frac{m_{VOC}*Q_L}{V*S} \tag{5}$$

where $E_{VOC}$ is the VOC Emission rate (ng m$^{-2}$ min$^{-1}$); $m_{VOC}$ is the mass of the VOC (ng), as determined by the thermal desorption calibration; $Q_L$ is the flow through the leaf chamber (L min$^{-1}$); $V$ is the total volume of air sampled with the sorbent tube (L), sampling flow multiplied by sampling time; and $S$ is the leaf surface area (m$^2$). For all measurements in this manuscript, we selected leaves that filled the chamber, for a total measured leaf area of 6 cm$^2$. However, this technique is still applicable for leaves that do not fill the chamber due to size or shape; for such leaves, leaf area must be determined separately (e.g., via image processing (Chaudhary et al., 2012), or via calculations based on geometric measurements (Sellin, 2000)).

**2.4 Sampling protocol**

The sampling protocol involves clamping the PPS leaf chamber onto a leaf, waiting for the leaf to adapt to the leaf chamber conditions, collecting trace gas measurements from the SAM and REF subsampling ports, and then either removing the leaf chamber and moving to a new leaf (single emission point), or changing the environmental conditions to investigate leaf-level emissions responses to temperature, light, relative humidity, or $CO_2$ (Table 1). Photosynthesis may be measured simultaneously at any point in the sampling protocol, and is independent of emission measurements.

Once the PPS has undergone its standard warmup (≤45 min), we set the PPS to the standard environmental conditions and allow the instrument to equilibrate without a leaf present, with the leaf chamber closed (<15 min; the further the ambient conditions deviate from standard conditions, the longer the instrument takes to equilibrate). We match the IRGAs to one another (LI-COR, 2017) prior to collecting an emissions measurement, when the $CO_2$ or humidity values change, or within an hour since the last match. To collect a system background ('system blank'), we connect a sorbent tube to the SAM subsampling port and use an external handheld pump to sample emissions (0.2 L min$^{-1}$; 20 minutes). The tube is then removed and the subsampling port capped. To sample leaf emissions, we enclose a leaf in the PPS chamber and allow the leaf to acclimate at standard conditions (30 seconds to 35 minutes). A sorbent tube and external pump connected to the SAM subsampling port samples the leaf emissions (0.2 L min$^{-1}$; 20 minutes). For CIMS measurements, we collect a system blank from the PPS (no leaf) by sampling the SAM subsample port for at least 5 min (at 1 Hz). After enclosing the leaf in the PPS chamber, we monitor the stability of both photosynthesis and volatile emissions. We typically find that the leaf and detector system requires at least 5 min to stabilize.

If there are no internal sources or sinks to the VOCs of interest (or these interactions are quantifiable), gas measurements may be simultaneously taken from both the REF and SAM subsample ports (with flow considerations, as described in Sect. 2.1). With this method, REF measurements provide the background for subtraction from the SAM emissions measurements.

At this point, users may make continuous measurements, survey measurements, or response measurements. A continuous measurement allows for the subsequent measurement of the same leaf tissue at the same environmental conditions (i.e. one leaf throughout the day). A survey measurement allows for the measurement of multiple leaves under one set of environmental conditions (i.e. sampling emissions from multiple leaves on the same plant). Importantly, each time a leaf is physically placed in the PPS chamber, it requires time (30 s – 35 min, depending on environmental conditions) to acclimate. A response measurement allows for the measurement of a single leaf at different environmental conditions (e.g. sampling emissions as a function of temperature).

Leaves must acclimate to new environmental conditions. However, the time required for a leaf to adapt to placement in the chamber or changing environmental conditions is inconsistently reported in leaf-level photosynthesis studies. Some studies allow leaves to acclimate until photosynthesis reaches stability or steady-state (e.g., Bunce, 2008; Domurath et al., 2012), though those terms are often undefined. Some studies use an upper (e.g., Yang et al., 2010) or a lower (e.g., Lang et al., 2013) time limit to allow photosynthesis to reach stability. When exact equilibration times are mentioned, they vary greatly between perturbations and between studies. For emissions measurements, equilibration times of both photosynthesis and BVOC emission must be considered. Using the CIMS, we determined that it takes 10-15 minutes for both photosynthesis and formic acid to reach stability after being clamped or after an environmental change.

We investigated the potential for VOCs in the leaf chamber to persist from one experiment to another, after the leaf has been removed, through adsorption on gaskets or chamber surfaces ("carryover"). Carryover can cause spuriously high emission measurements. To investigate carryover, we collected a system blank (no leaf present; SAM port) before introducing a ponderosa lemon (*Citrus limon x Citrus medica*) leaf into the chamber for the next 8 hours at varying temperatures. We observe no consistent evidence that longer periods in the leaf chamber impact photosynthesis or VOC emission. Citrus is believed to influence regional atmospheric chemistry due to their VOC emissions (Hodges and Spreen, 2006; Park et al., 2013). As a cocktail-sized, slow-growing plant with large leaves, this species was suitable for laboratory and greenhouse experiments. Immediately after removing the leaf at the end of the day, we collected a second system blank (no leaf present; SAM port).

We observed no carryover of monoterpenes (α-pinene, β-pinene, limonene, cis-β-ocimene, or γ-terpinene) or caryophyllene. The only identifiable plant emissions with observable signal (% of initial, i.e., leaf in chamber) that persisted after the leaf was removed were citral (27%) and 2-ethyl-1-hexanol (92%).

We also observed carryover of long-chain acids including palmitoleic acid (49%), pentadecanoic acid (47%), hexadecanoic acid (85%) and oleic acid (88%). Squalene (89%) also had substantial carryover. These compounds could have been introduced via the cuticular wax of leaves (Fernandes et al., 1964) or through human contact. However, these signals appear at retention times between 15 and 17 min, and are thus unlikely to interfere with signals of more volatile species.

Volatility likely plays a role in the carryover of compounds. Squalene, citral, and the long-chain acids have lower volatility than the monoterpenes. However, 2-ethyl-1-hexanol is of similar volatility to the monoterpenes, yet persists after the leaf has been removed. Carryover should thus be investigated for specific compounds prior to extensive studies.

## 2.5 Leaf chamber conditions

The PPS control of environmental conditions enables acquisition of short-term response curves for trace gas emissions, which are typically used to parameterize biogenic VOC emissions in atmospheric chemical transport models. Table 1 summarizes the ranges in parameters we find to be feasible for each environmental parameter.

The PPS regulates $CO_2$ and light well. However, both temperature and humidity regulation in the PPS depend on the balance between ambient and desired conditions. Relative humidity is constrained so as to not reach condensing conditions, so the extent of RH control depends on the temperature of the leaf chamber. For example, when aiming for high PPS temperatures (>30 °C), the PPS can have difficulty simultaneously maintaining high (>50 %) RH. When ambient temperatures are low (<4 °C), the PPS is challenged to maintain RH >35 %. This instrumental challenge occurs because temperature control in the PPS is limited by the heat exchanger; as the heat exchanger approaches dew point, the PPS takes proactive measures and

slows the heating or cooling of the system. We find two approaches to deal with PPS temperature/RH problems: (i) temperature may be set independently of humidity, or (ii) temperature may be ramped slowly while humidity is maintained.

Of all of the controllable environmental conditions, temperature takes the longest for the PPS to regulate ($10 \pm 2$ minutes to warm an empty chamber from 33 to 18 °C). Cooling takes twice as long as heating, and introducing a leaf into the chamber increases time necessary to cool by 35 % and time necessary to heat by 26 %. External fans improved the chamber temperature control at higher ambient temperatures, as did placing ice packs beside the air-inlet, around the chemical tubes, beside the leaf chamber, and on the side of the head improves the temperature control.

The LI-6800 also enables direct control of leaf vapour pressure deficit, but achieving a large dynamic range in vapour pressure deficit is subject to the same constraints as simultaneously changing temperature and RH in the PPS.

## 3 Internal PPS versus external $CO_2$ measurements

The LI-6800 PPS internally measures leaf-level $CO_2$ exchange with the SAM and REF IRGAs as a core measurement, providing $CO_2$ assimilation ($\mu$mol m$^{-2}$ s$^{-1}$). Assimilation provides a useful metric of validation against external leaf-level emissions, and we compare leaf-level $CO_2$ assimilation measured internally by the LI-6800 PPS and externally through the subsampling manifold and an external $CO_2$ analyzer. Here, we used the $CO_2$ assimilation of a basil leaf (*Ocimum basilicum*) to verify that the use of an external subsampling port supports the same values as the PPS's internal IRGA systems.

We connected an external $CO_2$ analyzer (LI-840A, LI-COR, Nebraska) to the PPS (no leaf) and varied the $CO_2$ mixing ratio to determine the sensitivity of external $CO_2$ measurements (using the LI-840A) with the internal LI-6800 $CO_2$ measurements. The LI-6800 can control $CO_2$ mixing ratio in one of two locations: before (REF) or in (SAM) the leaf chamber. First, we compare $CO_2$ measurements between the internal (LI-6800) and external (LI-840A) $CO_2$ analyzers. We internally controlled the REF $CO_2$ mixing ratio and measured the subsequent $CO_2$ mixing ratio externally though each subsampling port. We then controlled the SAM $CO_2$ mixing ratio and repeated the external measurements. All comparison experiments showed a strong correlation between internal and external $CO_2$ measurements ($R^2 > 0.9999$). The controlled $CO_2$ mixing ratio for both experiments ranged from 0 through 1600 ppm. We found the external $CO_2$ measurement was 5.5 % higher than the internal measurement, which we attribute to systematic differences in instrument calibration (Fig. 3). We find no evidence of leaks at below-ambient $CO_2$ mixing ratios.

We then compared $CO_2$ assimilation (sampling with leaf) between the internal PPS determination and the external measurements accounting for observed flows, etc. This external $CO_2$ assimilation measurement and calculation approach parallels our coupled PPS+online sampling trace gas measurement, and provides validation of the sampling approach. For

$CO_2$ assimilation comparisons, we controlled the SAM $CO_2$ mixing ratios and monitored the REF $CO_2$ mixing ratios externally. We accounted for the calibration offset between internal and external $CO_2$ detectors. With the external $CO_2$ analyzer connected to the REF subsampling port and a leaf in the chamber, we set the PPS $CO_2$ mixing ratio to 200, 400, 600, 800, and 1000 µmol $CO_2$ mol$^{-1}$. The PPS measured photosynthesis 10 times within 10 minutes while we externally monitored $CO_2$ mixing ratios from the REF port (1 Hz).

We calculate $CO_2$ assimilation (A) as:

$$A = \frac{Q_{L,c}*\left([CO_2]_R - [CO_2]_S*\frac{1-[H_2O]_{R,c}}{1-[H_2O]_{S,c}}\right)}{S} \qquad \text{(6, adapted from LI-COR, 2017)}$$

where $Q_{L,c}$ is the flow through the leaf chamber (µmol s$^{-1}$), multiplied by the leak correction factor (unitless, provided by the PPS); $[CO_2]_R$ and $[CO_2]_S$ is the mixing ratio of $CO_2$ (µmol mol$^{-1}$), as determined by the REF and SAM infrared gas analyzers, respectively; $[H_2O]_{R,c}$ and $[H_2O]_{S,c}$ are the mixing ratio of $H_2O$ (mol mol$^{-1}$), as determined by the REF and SAM infrared gas analyzers, respectively; S is the leaf area (m$^2$). We take $[H_2O]_{R,c}$ and $[H_2O]_{S,c}$ from the LI-6800.

The internally and externally calculated $CO_2$ assimilations correlate well ($r^2 = 0.97$) with 1 % difference between the two approaches (Fig. 3).

## 4 Trace gas backgrounds in the PPS

Background contamination reduces analyte signal accuracy. Co-eluting peaks in a gas chromatogram add additional difficulty in determining the exact peak area of a VOC analyte. When a chromatogram features heavy background contamination from a system, the chromatograms can become busy, challenging untargeted peak identification. Here we investigate the background VOCs in the PPS.

The REF port can be measured simultaneously with the SAM port to provide a background measurement of air entering the leaf chamber, but not any internal PPS sources of interferences in the leaf chamber.

The PPS is made of materials that can emit volatile compounds. While PPS system background may not contribute substantial background signals when using certain targeted analytical techniques (e.g., selected ion monitoring GC/MS), untargeted techniques, such as full scan GC/MS, are susceptible to background interference. TD-GC/MS chromatograms of the PPS (no leaf, 30 °C) revealed substantial background contamination, especially compared to the background of the Tenax tubes themselves (Fig. 4). The total integrated ion counts of identifiable peaks were 49 % higher background from the SAM port versus the REF, highlighting the problem of only using the REF port as a background for VOC analysis. These peak counts are substantially higher (by ~80 %) than the blank Tenax sorbent tube itself. Primary differences in the

integrated peak area between SAM and REF are due to the five largest peaks, three of which are siloxanes. Siloxanes are commonly used in consumer products, including textiles, cosmetics, paint, and electronics (Fromme, 2018; Tilley and Fry, 2015), and were 41 % higher in the SAM than REF ports. The other two largest peaks are isobornyl acrylate (a film-forming agent) and n-octyl acrylate (an adhesive and coating component). While unlikely to interfere with leaf VOC emissions, co-elution with these peaks may lead to unidentified emissions in untargeted approaches. As a result of this work, we recommend taking frequent backgrounds from the SAM port to ensure no chamber background interference for anaytes of interest.

Figure 5 categorizes the signals from the SAM background by functional groups, highlighting the complexity and potential interferences for biogenic trace gas emission analysis. The large background signals caution against using bulk signal measurements (e.g. total observed carbon, or total observed reactivity) from the PPS without careful background analysis. Instead, targeted approaches like extracted ion chromatography (EIC) are a promising way to exclude spurious background signals. Figure 6 highlights the differences between the full chromatogram (total ion counts) and an EIC, where we selected for monoterpenes. This approach clearly separates leaf-emissions that are not present in the blank, including β-pinene (4.210 min), limonene (5.175 min) and β-ocimene (5.561 min). By minimizing background contamination with EIC, we clearly observe differences between strongly- and weakly-emitting leaves (Fig. 6). Therefore, we recommend an EIC approach for the semi-targeted identification and analysis of monoterpenes and aldehydes.

We investigated three approaches to minimizing the PPS backgrounds. We replaced the Drierite dessicant with silica gel orange (Sigma-Aldrich, 13767-2.5KG-R) and the Stuttgarter Masse humidifier with Perlite (Miracle Gro®, 74278430). We also installed fresh air filters at each chemical column, IRGA and the air-inlet. After each change, we flushed the system with heated air (35 °C at a flow rate of 1300 μmol s$^{-1}$ for 30 minutes) before collecting system blanks under standard conditions, but none of these changes substantially decreased the background signals (Fig. S3).

The air entering the PPS is ambient, and thus prone to change throughout the day as sources and sinks vary. While the PPS includes several filters within the system, they do not filter all biogenic hydrocarbons – including monoterpenes. This is a particular problem in greenhouses, where low exchange rates, warm temperatures and large concentrations of plants lead to high ambient biogenic VOC emissions. We investigated the potential to filter monoterpenes from inlet air at the Plant Growth Facilities at Colorado State University. We added a home-built charcoal filter (30.5 cm piece of 85 cm o.d. stainless steel tubing filled with activated charcoal Norit® (Sigma-Aldrich, 29204-500G) and filtered with glass wool and stainless steel mesh on either end) to the air inlet of the PPS. This filter completely removed all background α-pinene from 0.04 ppb to below detection limit, but was less effective in subsequent outdoor experiments. As the ambient concentration of VOCs vary with time of day, we thus recommend both using a charcoal filter and taking simultaneous REF and SAM measurements to

account for interferences from input air. Alternately, zero air can replace ambient input air at the PPS inlet, per
       manufacturer's instructions (LI-COR, 2017).

## 5 Case studies

       Despite these background interferences, the LI-6800 has the potential to investigate plant gas exchange for an array of
       molecules with an array of trace gas instrumentation. Here, we provide case studies with both online (5.1; CIMS) and offline
(5.2, 5.3; TD-GC-MS) analysis. TD offers the benefit in that the sorbent tubes are easily portable, though sample collection
       and analysis is time-intensive. CIMS offers the benefit of online, real-time data acquisition, however the instrument itself is
       less portable and provides no definitive compound identities. These case studies maintained standard conditions unless
       otherwise noted, and each study used different plants. Further information on plant growth conditions can be found in Sect.
       S4.

### 395  5.1 Formic acid emissions

       Organic acids account for roughly 25 % of global non-methane VOCs (Khare et al., 1999) and contribute to secondary
       organic aerosol (Yatavelli et al., 2014). Despite their ubiquity, models typically underestimate ambient concentrations of
       formic acid, even the structurally simplest of organic acids, implying a missing source (Paulot et al., 2011; Alwe et al.,
       2019). This missing source of formic acid is not soils (Mielnik et al., 2018), but flux studies (Fulgham et al., 2019) and
vertical gradient measurements (Mattila et al., 2018) suggest a direct ecosystem source. Here we demonstrate the capacity of
       the PPS coupled to a CIMS system to investigate leaf-level organic acid sources from *Mentha spicata* (spearmint), a culinary
       herb of economic importance due to the production of its essential oil.

       We conducted a temperature response curve on a spearmint leaf connected to the PPS with a CIMS detector. We performed
three temperature response curve replicates, each with temperatures varying from 21 to 35 °C. The leaf was acclimated (as
       described in Sect. 2) at each new temperature for at least 5 minutes, during which time the CIMS sampled the REF port to
       determine the system background. We then simultaneously measured leaf-level emissions of formic acid and photosynthetic
       parameters for 5 minutes.

$CO_2$ assimilation and formic acid emission both varied with temperature for this leaf (Fig. 7). As temperature increases, $CO_2$
       assimilation increases up to a maximum value of 14.2 $\mu mol\ m^{-2}\ s^{-1}$ at 26 °C. This $CO_2$ assimilation follows the expected
       cubic fit (Yamori et al., 2010). In contrast, formic acid continues to increase above the photosynthesis maximum, with
       maximum emission (2.30 $\mu g\ m^{-2}\ min^{-1}$) occurring at 29 °C. However, we emphasize that this represents a single experiment
       using the CIMS to demonstrate the utility of coupling the CIMS to the PPS, rather than an extensive or replicated experiment
of formic acid emissions. Thus, these observations should be considered a case study, rather than emissions ratios to be used

in models. While the terpenoids of the essential oils of spearmint have been investigated (Delfine et al., 2005), stored and emitted compounds may differ. There is a need for studies focusing on the leaf-level emission rate of VOCs, including monoterpenes and formic acid. This case study does demonstrate the potential for the PPS to be coupled to real-time measurements in exploring less-studied BVOCs, such as organic acids, at a leaf level.

**5.2 Decanal emissions**

$C_6 – C_{10}$ aldehydes are an understudied class of plant BVOC emissions (Ciccioli et al., 1993; Wildt et al., 2003; Owen et al., 1997). Aldehydes can contribute to free radical formation in the atmosphere through photolysis or reaction with OH radicals (Atkinson, 1986). Decanal ($C_{10}$-aldehyde) is present in atmospheric mixing ratios of $ppt_v$ to $ppb_v$ (Ciccioli et al., 1993), and is emitted by plants in response to stress (Wildt et al., 2003). Here, we demonstrate the potential for off-line measurements
(i.e. the TD-GC/MS) coupled to the PPS to investigate plant emissions of $C_6$-$C_{10}$ aldehydes. Figure 8 shows the temperature response curve of a single leaf on a basil plant (*Ocimum basilicum*), a popular culinary herb. We collected single sorbent tubes for 20 minutes at each point as we varied temperature by ~3 °C from 18 to 35 °C. The LI-6800 simultaneously measured photosynthesis every 30 seconds. Background decanal concentrations in ambient air were 11 ± 1 (average ± standard deviation) parts-per-billion.


$CO_2$ assimilation increases over the entire range of temperatures, beginning to stabilize around 33 °C. The $CO_2$ assimilation values for this study are within range of values from previous studies (Golpayegani and Tilebeni, 2011). A cubic fit to assimilation suggests that 35 °C was the maxim in assimilation, which would decrease at higher temperatures. In contrast to photosynthesis, decanal exhibits bi-directional exchange. As temperature increases, decanal emissions are initially zero or
negative (i.e. lower than the background concentration of input air), and then show enhanced uptake with increasing temperature before a turn-over point at which emission rapidly increases. The temperature response is inconsistent with stored pools (Grote et al., 2013), suggesting a more complex biochemical pathway.

The observed uptake of decanal below 27 °C supports the idea of a turnover point and bidirectional exchange of VOCs
(Niinemets et al., 2014; Millet et al., 2018). Further investigation of turnover points as a function of varying input air VOC concentrations are warranted. Essential oil emissions of monoterpenes are quantified for basil (Tarchoune et al., 2013), however, the leaf-level emission of decanal is understudied, and has not yet been investigated for this species. The range of decanal emissions vary greatly in this study, but our findings suggest that, at high temperatures, decanal may be more strongly emitted than previously found. Our highest emissions at 35 °C are over 200 times greater than emission rates found
from canola plants (Wildt et al., 2003). There is need for further study investigating the interspecies differences in aldehyde emissions, in addition to the light and temperature dependencies of decanal emissions.

Temperature response of photosynthetic metrics can be used to compare the thermotolerance between species or between plants of the same species. For example, this study suggests that basil has a photosynthetic maximum at temperatures greater than spearmint (35 °C, Fig. 8;26 °C Fig. 7, respectively), despite the fact that basil had a lower overall $CO_2$ assimilation rate. At temperatures above the maximum, photosynthesis and plant productivity may be inhibited (Berry and Bjorkman, 1980), suggesting that basil may have a higher thermotolerance than spearmint. We note that this comparison only considers short-term temperature increases, and further investigations would be necessary to determine the acclimation potential of these plant species to higher temperatures. Temperature response of trace gases can be used to further investigate the mechanisms by which different compounds are emitted. Comparing the emission of lesser-studied compounds like decanal to better-studied compounds like monoterpenes can improve the understanding of the regulating factors in leaf-level BVOC emissions.

### 5.3 Monoterpene emissions

The PPS-coupled emission sampling method is portable, which we take advantage of in our third case study. While BVOC emission studies often quantify emissions in terms of dry leaf weight, *in situ* measurements enable us to collect data based on leaf area, which is used in many emissions models.

To investigate the difference in limonene and γ-terpinene emissions between plants of different species, we sampled two shaded leaves of each of three tree species during the summer of 2019 in the Colorado State University Arboretum in Fort Collins, CO. We sampled: *Ginkgo biloba* (ginkgo), *Morus alba* (mulberry), and *Juglans regia* (walnut). These species cover a variety of uses: ginkgo is one of the longest living tree species and is used in dietary supplements (Strømgaard and Nakanishi, 2004), mulberry is a primary food source for silkworms and is used for paper production (He et al., 2013), and walnut is of economic importance as timber (Ares and Brauer, 2004). These three species are considered low emitters of monoterpenes (Benjamin and Winer, 1998); our identification and quantification of their monoterpene emissions highlight the sensitivity of this technique.

Emissions were taken at 27±2 °C, near-ambient $CO_2$ (414 ppm), and under saturating light conditions (2000 μmol m$^{-2}$ s$^{-1}$). We simultaneously sampled monoterpene emissions using the sorbent cartridges (30 minute collection) and photosynthesis (30 second time resolution) at each temperature. Leaf temperature was difficult to regulate in the field. The PPS maintained a 25 °C leaf temperature with ambient temperatures up to 29 °C, but could not keep leaf temperatures below 28 °C when ambient temperatures increased, even with shading and ice packs.

Previous studies have identified monoterpene emissions in ginkgo (Li et al., 2009), mulberry (Papiez et al., 2009), and walnut (Casado et al., 2008); however, these studies calculate emission rates in units of dry weight. Models that rely on leaf

area to calculate monoterpene fluxes must thus account for differences between dry weight and leaf area. Alternatively, emissions collected via this method are already normalized to surface area, and do not require a major conversion.

Here, limonene emissions from all species were an order of magnitude greater than γ-terpinene, by factors of 10-20 (Fig. 9). This ratio can change based on genotype; for example, the ratio of limonene to γ-terpinene emissions in different black

walnut genotypes range from 4.1:1 to 1:1.7 (Blood et al., 2018). Monoterpene emission rates from individual leaves varied, though this variance was more notable for γ-terpinene than limonene, in agreement with previous studies (Blood et al., 2018). For example, we found that limonene emission rates differed by 24 % between the two mulberry leaves, whereas γ-terpinene differed by 46 %.

Within leaves of a single plant, chamber temperature and subsequent $CO_2$ assimilation rates were similar (<0.5 % difference in assimilation between leaves of the same plant), and observed $CO_2$ assimilation rates agreed with previous measurements (Pandey et al., 2003; Baraldi et al., 2019; Nicodemus et al., 2008). This discrepancy in variance between $CO_2$ assimilation and monoterpene emissions on a single plant highlights the limitation of tying modelled photosynthesis rates to VOC emissions and warrants further investigation. We focus on two monoterpenes here, however, this field survey approach to

trace gas VOC emissions can provide a species-specific monoterpene emission cassette.

We provide an example monoterpene emission cassette. Figure 10 puts those emissions into atmospheric context. We show that, although α-pinene contributes to 22% of the measured emissions, it only contributes to 7% of overall OH formation and 0.5% of ozone formation. Although α-terpinene contributes to 20% of the measured emissions, it is the dominating factor in

both OH and ozone formation (44% and 98%, respectively). This technique allows for the speciation necessary to understand both the factors which influence emission rates and their subsequent atmospheric impact.

This case study supports previous findings that leaf emissions can vary between leaves of one tree (Staudt et al., 2001b), between trees of one species (Staudt et al., 2001b), and between trees of different species (Benjamin et al., 1996) – but that

trace gas sampling with the PPS is a viable method for investigating these sources of variance. We further highlight the importance of speciated monoterpene analysis, and this technique's application for such analyses.

## 6 Conclusions

This study shows the utility of a new PPS system coupled with both on- and off-line analysis for the analysis of leaf-level gas emissions, and the limitation and caveats associated with those measurements. In particular, trace gas measurements with high air flow needs (> 1 L min$^{-1}$) must be used carefully. Using an external $CO_2$ monitor to calculate $CO_2$ assimilation rates,

we verify the integrity of the subsampling manifold and provide relevant equations for calculations of plant gas exchange.

The PPS-coupling system described herein has substantial potential for improving our understanding of plant emissions. For example, different CIMS ionization sources can target different types of organic molecules (e.g. acetate ionization for organic acids vs iodide ionization for oxygenated organics), and different sorbent materials in thermal desorption tubes enable detection of different compounds (i.e. Tenax for monoterpenes vs graphitic carbon for isoprene). However, we emphasize the importance of carefully considering potential contaminants from the PPS itself, and the use of frequent system background measurements through both the SAM port in the absence of a leaf, and the REF port in presence of the leaf. The further potential to control the composition of the airflow into the PPS will enable investigation of compensation points.

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

**Table 1. User-defined standard, tested and operating conditions of environmental controls using the LI-6800.**

| | Chamber flow ($\mu mol\ s^{-1}$) | Chamber overpressure (kPa) | Fan speed (rpm) | Relative humidity (%) | Photon Flux Density[c] ($\mu mol\ m^{-2}\ s^{-1}$) | Temperature (°C) | $CO_2$ ($\mu mol\ mol^{-1}$) |
|---|---|---|---|---|---|---|---|
| Standard conditions | 500 | 0.1 | 10,000 | 50 | 750 | 25 | 400 |
| Tested conditions [a] | 0 – 1475 | 0.0 – 0.2 | 3,000 – 14,000 | 0 – 75% | 0 – 3000 | 10 – 38 | 0 – 2000 |
| Operating conditions [b] | 0 – 1400[d] | 0.0 – 0.2 | 10,000 | 0 – 90% | 0 – 3000 | ± 10 from ambient | 0 – 2000[e] |

[a] Provided values indicate the range at which the instrument functioned properly in conditions tested at 1.5 km above sea level, ~0.84 atm (8.6 kPa).

[b] Recommended operating values from (LI-COR, 2017).

[c] Saturating light conditions recommended for most uses. Operating range dependent on temperature, values shown are for 25 °C.

[d] At standard ambient temperature (25 °C) and pressure (100 kPa, 0.99 atm).

[e] Exact values limited on bulk flow rate, review (LI-COR, 2017) for further details.

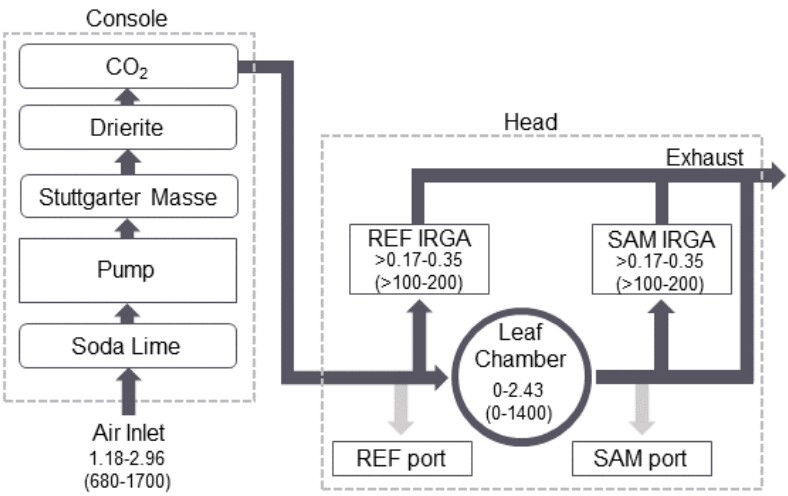

**Figure 1. Flow chart diagram of air flow through the PPS during emissions sampling. Dashed lines delineate flow through the PPS console and the head. Dark grey lines show the flow through the PPS during photosynthesis measurements. Light grey lines indicate the additional flow path during emissions sampling. Values for flow rate are given in L min$^{-1}$, with µmol s$^{-1}$ in parentheses. The order of the chemical treatment of air is shown for the console.**

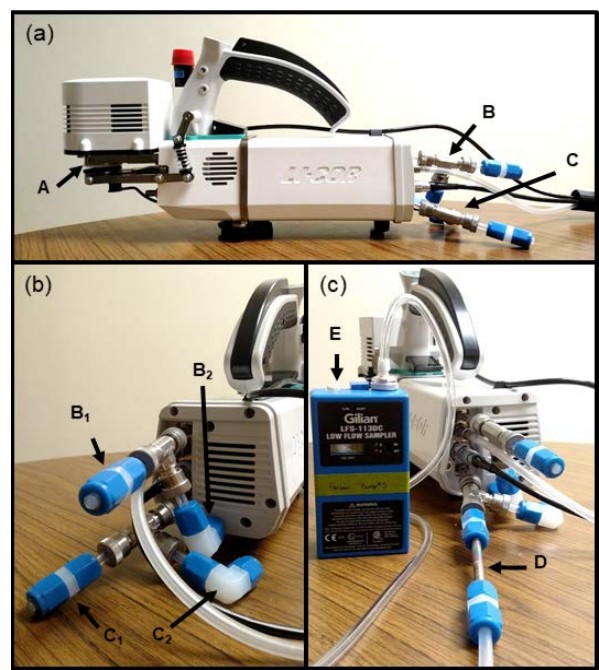

Figure 2. Photograph of the emissions subsampling manifold for the LI-6800. The profile view (a) highlights the leaf chamber (A), SAM subsampling port (B) and REF subsampling port (C). The back view (b), highlights the SAM and REF sampling ports ($B_1$ and $C_1$, respectively) and overflow ports ($B_2$ and $C_2$, respectively). Panel (c) shows an example setup of sorbent tube (D) emission collection with an external pump (E) sampling the REF subsampling port.

790

Table 2. Summary of monoterpenes quantified using TD GC/MS.

| Compound ($C_{10}H_{16}$) | RT [a] (min) | RSD [b] (%) | LOD [c] (ng) | Emission Rate LOD [d] (ng m$^{-2}$ min$^{-1}$) |
|---|---|---|---|---|
| α-pinene | $3.716 \pm 0.008$ | 8.2 | 0.137 | 11.4 |
| β-pinene | $4.44 \pm 0.01$ | 7.4 | 0.082 | 6.8 |
| α-terpinene | $5.173 \pm 0.008$ | 4.5 | 0.071 | 5.9 |
| p-cymene | $5.354 \pm 0.009$ | 5.6 | 0.111 | 9.2 |
| d-limonene | $5.47 \pm 0.01$ | 3.5 | 0.054 | 4.5 |
| γ-terpinene | $6.306 \pm 0.009$ | 2.4 | 0.085 | 7.1 |
| terpinolene | $7.35 \pm 0.01$ | 2.6 | 0.050 | 4.2 |

[a] Retention time

[b] Relative standard deviation (n=10)

[c] Limit of detection, calculated using the propagation of errors approach (Bernal, 2014)

795   [d] Based on a 20 min sampling time

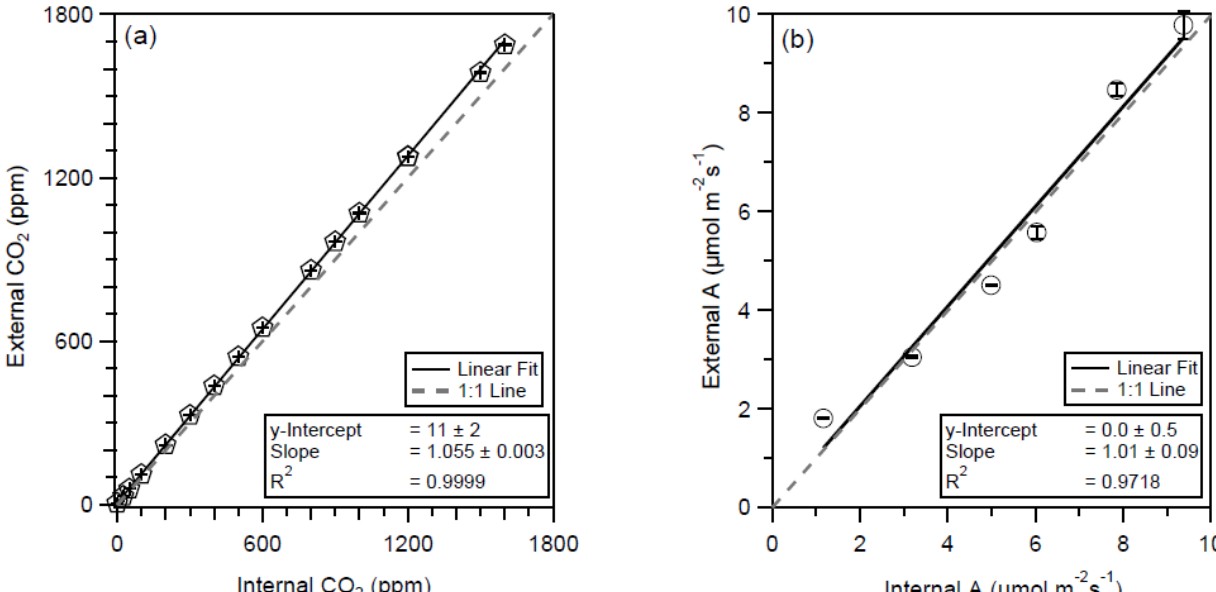

**Figure 3.** Correlation plots of $CO_2$ mixing ratio (pentagons, left panel) and $CO_2$ assimilation (A, circles, right panel) as calculated internally by the PPS (x-axis) and externally by the $CO_2$ analyzer (y-axis). A 1:1 line is present as a grey, dashed line. Linear regression fit is shown with a solid line, and fit parameters accompany in text, ± standard error of the fit. Error bars represent the standard deviation of values.

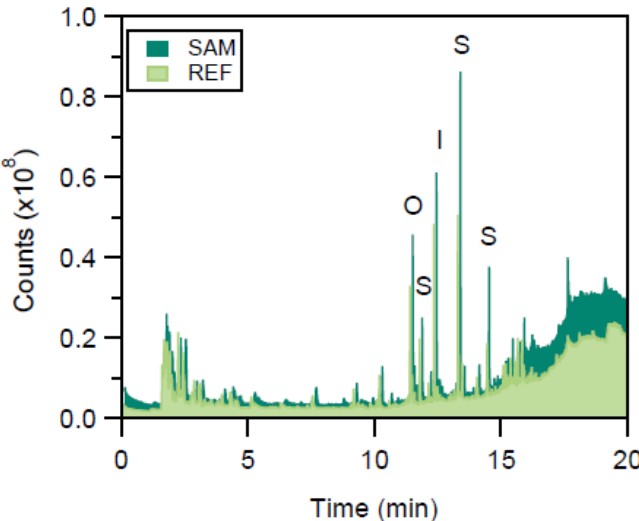

**Figure 4.** Stacked chromatograms of the background composition of the LI-6800, comparing measurements taken from the REF (light green) and SAM (dark green) ports as sampled by TD-GC-MS (20 minutes at 0.2 L min$^{-1}$, sampled on Tenax cartridges). The five largest peaks are labeled: S is the result of a siloxane, I is that of isobornyl acrylate and O is of n-octyl acrylate.

810

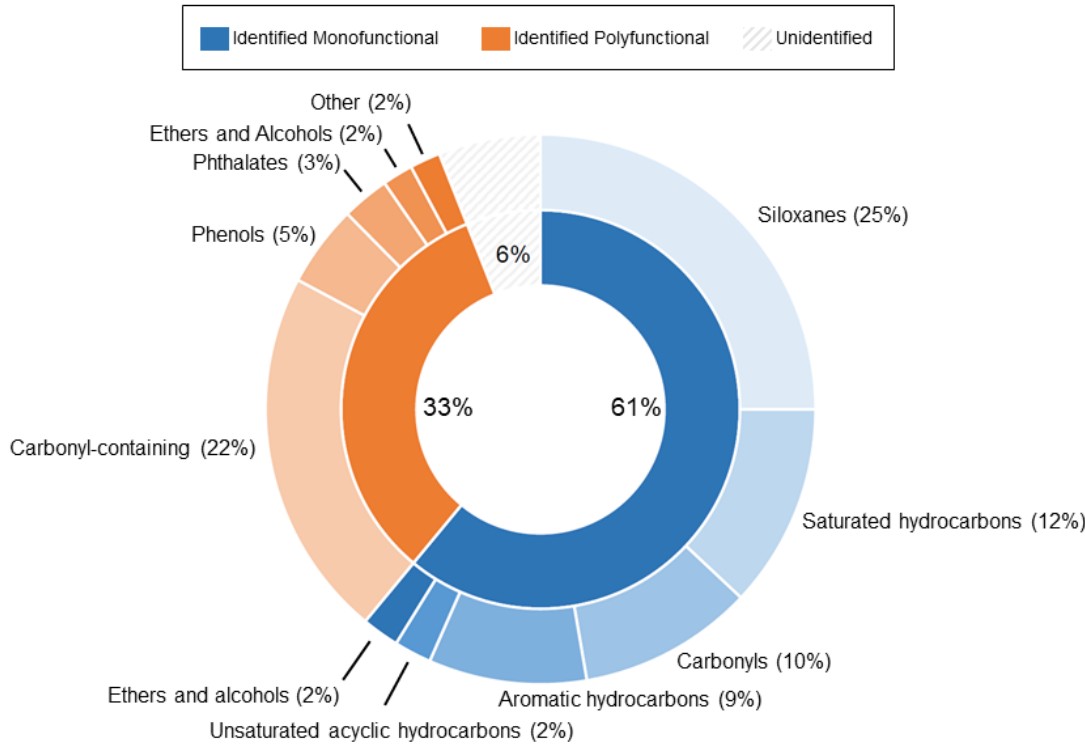

**Figure 5. Pie chart summarizing the background composition of the LI-6800 with no leaf in the chamber, collected using the SAM port of the PPS. Percentages are provided to indicate the contribution of each class of compounds to the total integrated peak area. The inner pie chart shows the division of total ion counts for identified monofunctional (containing a single functional group), identified polyfunctional (containing multiple functional groups) and unidentified (yellow stripes) peaks. Identification required an integrated peak area over 50,000 counts and a NIST library match score of at least 500. The outer pie chart shows the subsequent breakdown of both identified classes.**

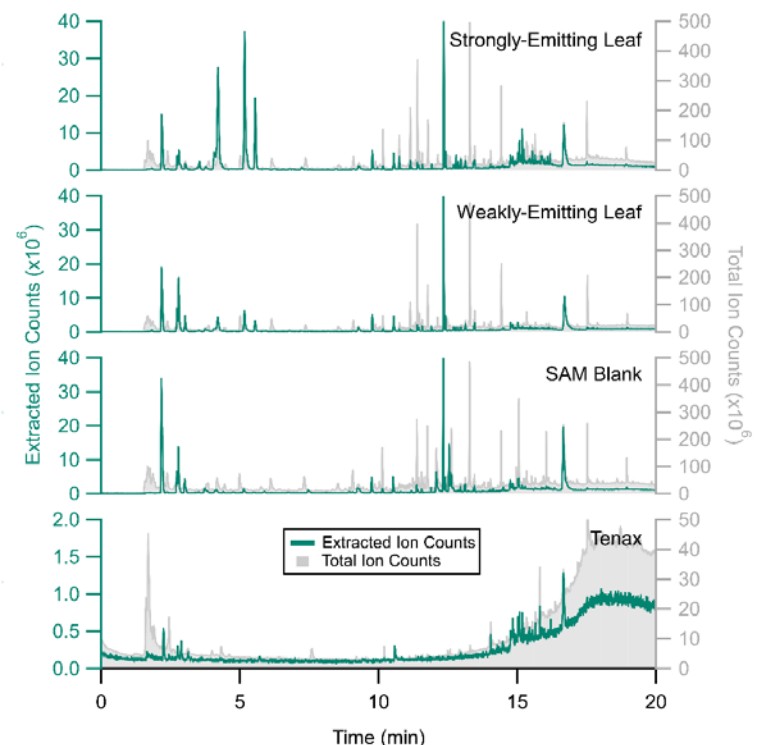

**Figure 6. Stacked chromatograms comparing extracted ion chromatograms (EIC using *m/z* 136, 135, 93 and 91, left, in green) with the total ion chromatogram (TIC, right, in grey) for a Tenax blank, a SAM blank, a weakly- and a strongly- emitting citrus leaf (*Citrus limon* × *Citrus medica*). Note the difference in axis scales between EIC and TIC. While several background peaks remain in the EIC, there are substantially fewer in the 10+ minute range. Peak height of EIC isobornyl acrylate (RT = 12.3 min) in has been truncated. Note that retention times differ from Table 2; the column length had been shortened by the time of these measurements.**

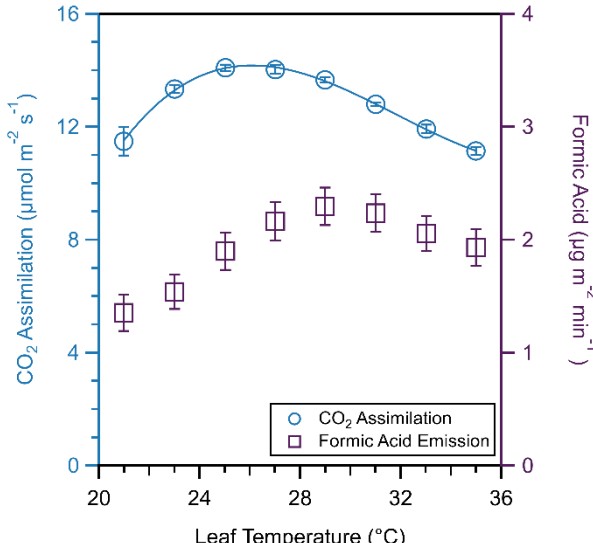

**Figure 7. CO$_2$ assimilation (blue circles) and formic acid emission (purple squares) temperature response curve of one spearmint leaf. Temperatures varied by 2 °C from 21 to 35 °C. CO$_2$ assimilation follows the expected cubic fit. We collected assimilation and formic acid emission measurements for five minutes and averaged the values of each; error bars represent the standard deviation of those averages.**

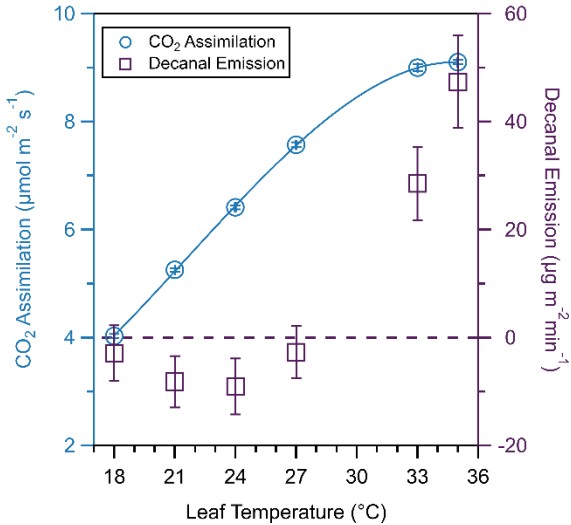

**Figure 8. CO$_2$ assimilation (blue circles) and decanal emission (purple squares) temperature response curve of one basil leaf. CO$_2$ assimilation is fit to a cubic function. We collected CO$_2$ assimilation values ten times over 20 minutes and averaged the values; error bars represent the standard deviation of those measurements. Error bars for decanal emission represent instrumental error. The dashed line denotes 0 µg m$^{-2}$ min$^{-1}$ decanal emission.**

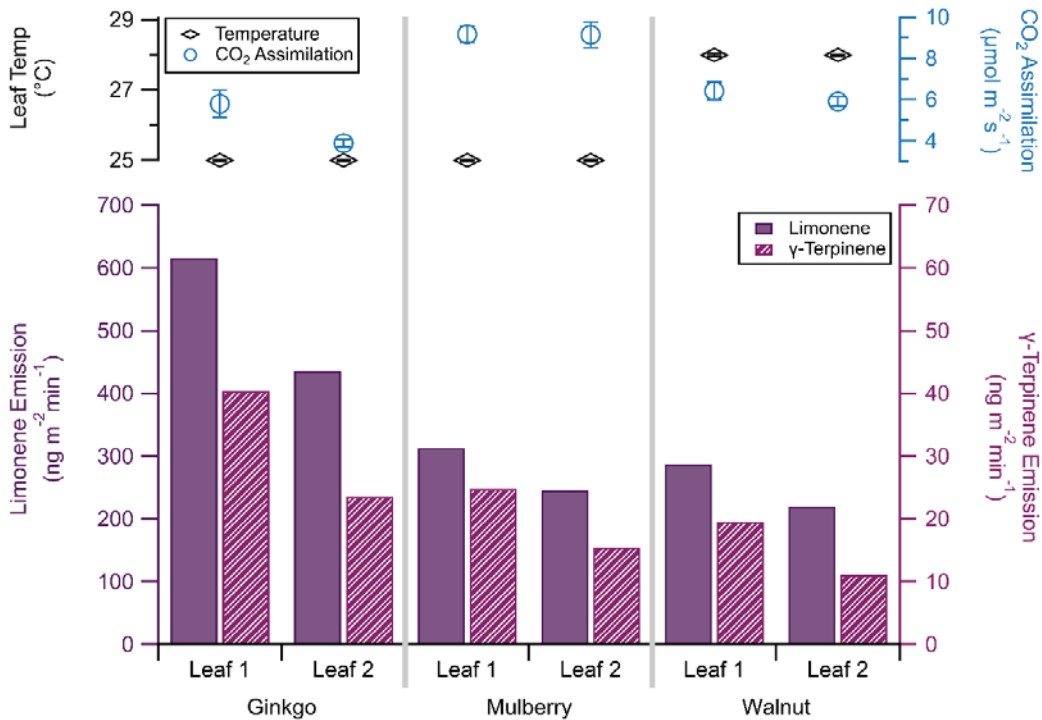

840 **Figure 9. Limonene (solid bars, left bottom axis) and γ-terpinene (striped bars, right bottom axis) emission of two leaves from each of three plant species: ginkgo, mulberry, and walnut. Note that the scale of the limonene emission axis is ten times that of the γ-terpinene emission axis. Leaf temperature (black diamonds, left top axis) and $CO_2$ assimilation (blue circles, right top axis) are included, with standard deviation bars (n = 60).**

845

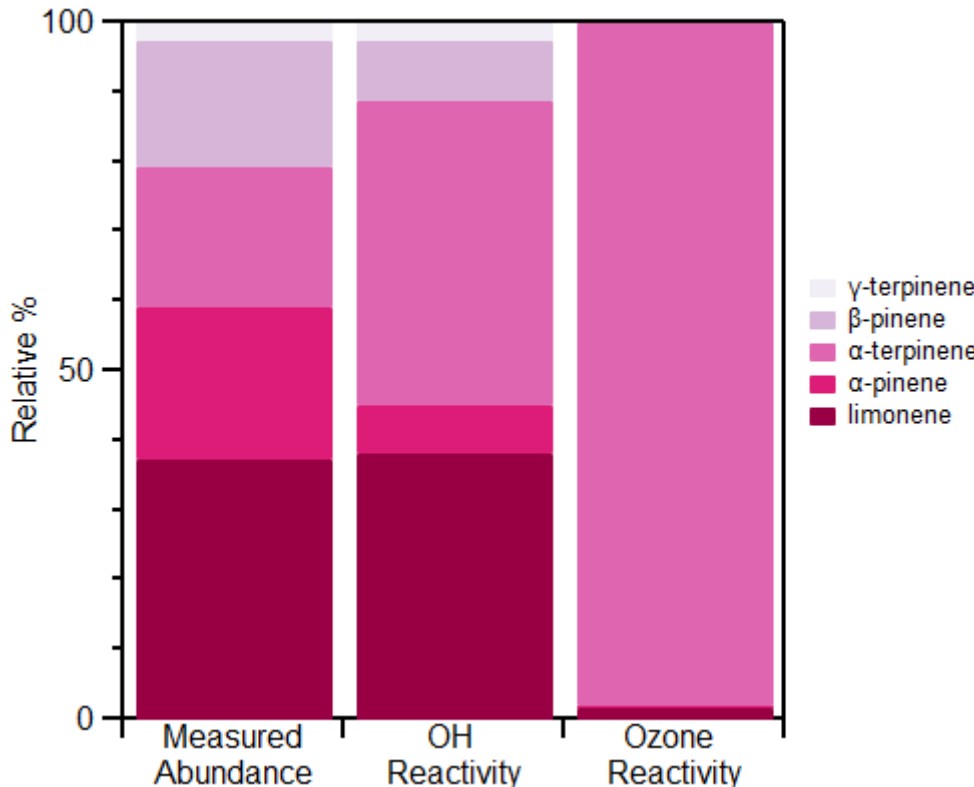

**Figure 10. Relative measured abundance of all quantified monoterpenes from Ginkgo Leaf 1 (Fig. 9), and the subsequent relative contribution to both OH Reactivity and Ozone Reactivity. OH and ozone reactivity were calculated using $k_{OH}$ and $k_{ozone}$ rate constants from Atkinson (1997).**

850

855

## Acknowledgements

We thank Dr. Karolien Denef for assistance with the GC/MS in the Central Instrument Facility CORE at Colorado State University and Elizabeth Gordon (Li-Cor, Inc.). We thank James Mattila for his assistance with CIMS calculations, and Tyson Berg and Jarod Snook for their assistance with data collection.

## Data Availability

Data presented in figures herein may be accessed online at
https://osf.io/8cs75/?view_only=9fddf94a16e94fd2a50289e01717ec65.

## Author Contribution

MR, DL, and DF designed the experiments and MR and DL carried them out. MR analysed the data in consultation with DL and DF. MR and DF prepared the manuscript.

## Competing Interests

The authors declare that they have no conflict of interest.