# Peer review of "Simultaneous leaf-level measurement of trace gas emissions and photosynthesis with a portable photosynthesis system"

_Atmospheric Measurement Techniques, 2020_

## Referee Comment (RC1) · Anonymous Referee #1 · 11 Apr 2020

**Summary and Recommendation**

Riches et al. present a method characterization technique that combines established methods in plant physiology for measuring photosynthesis with established off-line and on-line trace gas measurements. They couple a Licor, Inc. portable photosynthesis system (LI-6800) with two different approaches for measuring plant volatile emissions: adsorbent cartridge sampling followed by off-line thermodesorption GC-MS analysis and on-line trace gas monitoring with a ToF-CIMS using iodide ionization. The paper outlines the approach they used to couple these measurements, characterizes the chemical background of the measurement approach, and present a couple example "case studies" investigating plant volatile emissions as a function of temperature. They also validate the method by comparing $CO_2$ assimilation measured with the LI-6800 directly with $CO_2$ assimilation measured with an external $CO_2$ gas analyzer. The measurement approach and characterization are novel and would help the community address important questions about changing volatile emissions in a changing climate. The paper could be drastically improved with a few moderate revisions focusing on 1) better placing their results in context with other published results and 2) a more thorough analysis of the speciated monoterpene emissions which is data they have but was not presented. These concerns are described in more detail below. If these items are addressed, the paper would more adequately showcase the viability and utility of the measurement approach for addressing some of the major challenges in the field, including a better understanding of climate change effects on emission rates of different monoterpene isomers. I recommend publication after these moderate revisions.

MAJOR COMMENTS

Authors should indicate somewhere what the rationale was for selecting the plants they used in this study. This discussion should also include information on what is already known about their emissions, and how representative/relevant these plants might be overall. This will help place their results in context with the scientific literature on plant volatile emissions.

L. 245-251: results showing persistent signal of some compounds and not others are interesting and worthy of presentation in a results section rather than a methods section, particularly since this is a measurement technique development/characterization paper. I would like to see a more detailed presentation of the results from the carryover testing.

Section 5.1: Please place the $CO_2$ assimilation rates and volatile emission measurements in context with previous studies. Are these high values for plants overall? Low values? Typical for this type of plant in particular?

Section 5.3: It is unclear why the authors only present data on two of the monoter-penes they observed. Did all the monoterpene emissions have large variation between leaves and between trees? They have the data, and a more comprehensive analysis of that data is necessary to demonstrate the capability of this measurement approach for conducting the type of comprehensive, speciated terpenoid analysis that the com-munity needs to better understand factors regulating VOC emission rates from plants. Just showing results for two monoterpenes does not accomplish this.

MINOR COMMENTS

Regarding emission rate calculation (equation 5, L. 205): how are they estimating leaf surface area? There are a number of different approaches that often vary depending on type of plant. This should be clarified.

L. 364-365: "though sample collection and analysis is timely." Timely means "oppor-tune" or "prompt". I think the author intends to say something more like time-intensive or labor-intensive.

L. 411: "Temperature response curves can be used". Temperature "response" of what variable? Assimilation? trace gas exchange? All of the above? Unclear what is being referred to.

L. 412-413: "For example, this study suggests that basil has a higher photosynthetic thermotolerance than mint despite the fact that basil had a lower $CO_2$ assimilation rate." How so? Can you please elaborate on this referring to specific data in the plots that supports this conclusion?

L. 414: "to that of formic acid or monoterpenes can better inform of the impact and deciding factors". 2 comments. There is a typo or grammatical error around "can better inform of the impact". Also, "deciding" doesn't appear to be an appropriate word choice here. Perhaps "regulating"?

L. 437: "These case study data support that leaf emissions". Typo or grammatical
error?

Figure 8: decanal emissions, markers and error bars. If you don't have duplicates to define error bars as standard deviation, I suggest using your analytical error for the error bars. There should be some indication of error, even if it's the small error introduced from the instrumental analysis.

———————————————————

---

## Referee Comment (RC2) · Anonymous Referee #2 · 22 May 2020

General comments

Riches et al. present photosynthesis and trace gas (volatile organic compound, VOC) measurements made using a portable photosynthesis system (PPS) together with chemical ionisation mass spectrometry (CIMS) and offline analysis of sorbent tubes. PPS has been widely used to sample VOCs at well-defined photosynthetic conditions and this manuscript provides a useful characterisation of the LI-COR LI-6800 both in terms of $CO_2$ assimilation and measurement of VOCs. I have a number of minor concerns (listed below), with these points addressed I would recommend publication.

Specific comments

[Figure]

Concentration is used throughout when referring to mol mol-1, this should be mole fraction or mixing ratio.

Different units for emission factor are used when reporting results from the TOF-CIMS and sorbent tubes (moles and mass). It would be easier for the reader if the same units were used throughout.

Line 142. Brass fitting are often avoided when working with VOCs, why was a brass hose barb fitting used when all other fittings were stainless steel?

Line 147: Is the external pump placed up or downstream of the sorbent tube?

Line 165: The authors state that the impact of increased flow rates should be investigated for individual species. As a range of species were used in this study, was any effect observed?

Line 191: Most of the TD-GC/MS instrumental methodology is provided in the supplementary information which keeps the manuscript focused on the PPS system. On the whole I appreciate this but would like to see a make and model for the TD-GC/MS in the main text.

Line 209: The sampling protocol set out in section 2.4 focuses mainly on sampling to sorbent tubes. How did this differ for the CIMS?

Line 245: Did the authors observe any effect on photosynthesis or VOC emission when keeping a leaf in the chamber for long periods e.g. 8h?

Line 268: Does the use of external fans on the PPS impact the rest of the plant?

Line 310: Is there a temperature dependence in the trace gas background?

Line 365: "timely" is a vague term, could you estimate a time?

Line 403: What were the background decanal concentrations? Was this ambient air or was a VOC scrubber used as recommended in section 4?

[Figure]

Line 361: The authors describe the observations made in section 5 as a case study rather than emissions ratios to be used in models. While this is true I think that if a more detailed description of the plants used was provided (e.g. growth conditions, developmental stage etc.) readers would be able to make more use of the data presented. Perhaps this could be added to the supplementary information.

Technical corrections

Figure 3: caption refers to "squares, left panel" but the figure shows pentagons
* * *

---

## Author Comment (AC1) · 23 Jun 2020

**Reply to Anonymous Referee #1**

We thank the reviewer for their thoughtful and constructive comments. We have addressed all comments below, and we believe that the manuscript has improved with their suggestions. Reviewer comments are below (grey italicized), with author responses indented. Changes and additions to the manuscript are included, in bold.

*Riches et al. present a method characterization technique that combines established methods in plant physiology for measuring photosynthesis with established off-line and on-line trace gas measurements. They couple a Licor, Inc. portable photosynthesis system (LI-6800) with two different approaches for measuring plant volatile emissions: adsorbent cartridge sampling followed by off-line thermodesorption GC-MS analysis and on-line trace gas monitoring with a ToF-CIMS using iodide ionization. The paper outlines the approach they used to couple these measurements, characterizes the chemical background of the measurement approach, and present a couple example "case studies" investigating plant volatile emissions as a function of temperature. They also validate the method by comparing CO2 assimilation measured with the LI-6800 directly with CO2 assimilation measured with an external CO2 gas analyzer. The measurement approach and characterization are novel and would help the community address important questions about changing volatile emissions in a changing climate. The paper could be drastically improved with a few moderate revisions focusing on 1) better placing their results in context with other published results and 2) a more thorough analysis of the speciated monoterpene emissions which is data they have but was not presented. These concerns are described in more detail below. If these items are addressed, the paper would more adequately showcase the viability and utility of the measurement approach for addressing some of the major challenges in the field, including a better understanding of climate change effects on emission rates of different monoterpene isomers. I recommend publication after these moderate revisions.*

> We appreciate the reviewer's thorough involvement with the improvement of the manuscript, and their understanding of the importance of a thorough characterization of this measurement approach. We are pleased to address their concerns and comments below.

*MAJOR COMMENTS*

*Major1. Authors should indicate somewhere what the rationale was for selecting the plants they used in this study. This discussion should also include information on what is already known about their emissions, and how representative/relevant these plants might be overall. This will help place their results in context with the scientific literature on plant volatile emissions.*

> We have expanded upon our rationale and the literature surrounding our plants of interest. Additional details added are described in response to Major3, below.

> We have added to the final paragraph of Sec. 2.4: "**Citrus is believed to influence regional atmospheric chemistry due to their VOC emissions (Hodges and Spreen, 2006; Park et al., 2013). As a cocktail-sized, slow-growing plant with large leaves, this species was suitable for laboratory and greenhouse experiments.**"

We have added to the first paragraph of Sec. 5.1: "PPS coupled to a CIMS system to investigate leaf-level organic acid sources **from *Mentha spicata* (spearmint), a culinary herb of economic importance due to the production of its essential oil**."

We have added to the last paragraph of Sec. 5.1: "Thus, these observations should be considered a case study, rather than emissions ratios to be used in models. **While the terpenoids of the essential oils of spearmint have been investigated (Delfine et al., 2005), stored and emitted compounds may differ. There is a need for studies focusing on the leaf-level emission rate of VOCs, including monoterpenes and formic acid.**"

We have added to the first paragraph of Sec. 5.2.: "Figure 8 shows the temperature response curve of a single leaf on a basil plant (*Ocimum basilicum*), **a popular culinary herb**."

We have added to the second paragraph of Sec. 5.3.: "We sampled: *Ginkgo biloba* (ginkgo), *Morus alba* (mulberry), and *Juglans regia* (walnut). **These species cover a variety of uses: ginkgo is one of the longest living tree species and is used in dietary supplements (Strømgaard and Nakanishi, 2004), mulberry is a primary food source for silkworms and is used for paper production (He et al., 2013), and walnut is of economic importance as timber (Ares and Brauer, 2004). These three species are considered low emitters of monoterpenes (Benjamin and Winer, 1998); our identification and quantification of their monoterpene emissions highlight the sensitivity of this technique.**"

*Major2. L. 245-251: results showing persistent signal of some compounds and not others are interesting and worthy of presentation in a results section rather than a methods section, particularly since this is a measurement technique development/characterization paper. I would like to see a more detailed presentation of the results from the carryover testing.*

We chose to keep the carryover discussion in Sec. 2.4 to maintain the flow of the manuscript. However, we have replaced the original paragraph summarizing carryover (last paragraph of Sec. 2.4) and have expanded upon carryover results:

"**We observed no carryover of monoterpenes (α-pinene, β-pinene, limonene, cis-β-ocimene, or γ-terpinene) or caryophyllene. The only identifiable plant emissions with observable signal (% of initial, i.e., leaf in chamber) that persisted after the leaf was removed were citral (27%) and 2-ethyl-1-hexanol (92%).**

**We also observed carryover of long-chain acids including palmitoleic acid (49%), pentadecanoic acid (47%), hexadecanoic acid (85%) and oleic acid (88%). Squalene (89%) also had substantial carryover. These compounds could have been introduced via the cuticular wax of leaves (Fernandes et al., 1964) or through human contact. However, these signals appear at retention times between 15 and 17 min, and are thus unlikely to interfere with signals of more volatile species.**

**Volatility likely plays a role in the carryover of compounds. Squalene, citral, and the long-chain acids are lower volatility than the monoterpenes. However, 2-ethyl-1-hexanol is of similar volatility to the monoterpenes, yet persists after the leaf has been removed. Carryover should thus be investigated for specific compounds prior to extensive studies.**"

*Major3. Section 5.1: Please place the CO2 assimilation rates and volatile emission measurements in context with previous studies. Are these high values for plants overall? Low values? Typical for this type of plant in particular?*

We have expanded upon our rationale and the literature surrounding our plants of interest. Additional details added are described in response to Major1, above.

We have added to the second paragraph of Sec. 5.2.: "**The CO$_2$ assimilation values for this study are within range of values from previous studies (Golpayegani and Tilebeni, 2011).**"

We have added to the third paragraph of Sec. 5.2.: "**Essential oil emissions of monoterpenes are quantified for basil (Tarchoune et al., 2013), however, the leaf-level emission of decanal is understudied, and has not yet been investigated for this species. The range of decanal emissions vary greatly in this study, but our findings suggest that, at high temperatures, decanal may be more strongly emitted than previously found. Our highest emissions at 35 °C are over 200 times greater than emission rates found from canola plants (Wildt et al., 2003). There is need for further study investigating the interspecies differences in aldehyde emissions, in addition to the light and temperature dependencies of decanal emissions.**"

We have restructured Sec. 5.3. to include: "**Previous studies have identified monoterpene emissions in ginkgo (Li et al., 2009), mulberry (Papiez et al., 2009), and walnut (Casado et al., 2008); however, these studies calculate emission rates in units of dry weight. Models that rely on leaf area to calculate monoterpene fluxes must thus account for differences between dry weight and leaf area. Alternatively, emissions collected via this method are already normalized to surface area, and do not require a major conversion.**

Here, limonene emissions from all species were an order of magnitude greater than γ-terpinene, by factors of 10-20 (Fig. 9). **This ratio can change based on genotype; for example, the ratio of limonene to γ-terpinene emissions in different black walnut genotypes range from 4.1:1 to 1:1.7 (Blood et al., 2018).** Monoterpene emission rates from individual leaves varied, though this variance was more notable for γ-terpinene than limonene, **in agreement with previous studies (Blood et al., 2018)**. For example, we found that limonene emission rates differed by 24 % between the two mulberry leaves, whereas γ-terpinene differed by 46 %.

Within leaves of a single plant, chamber temperature and subsequent CO$_2$ assimilation rates were similar (<0.5 % difference in assimilation between leaves of the same plant), **and observed CO$_2$ assimilation rates agreed with previous measurements (Pandey et al., 2003; Baraldi et al., 2019; Nicodemus et al., 2008)**. This discrepancy in variance between CO$_2$ assimilation and monoterpene emissions on a single plant highlights the limitation of tying modelled photosynthesis rates to VOC emissions and warrants further investigation. **We focus on two monoterpenes here**, however, this field survey approach to trace gas VOC emissions can provide a species-specific monoterpene emission cassette."

*Major 4. Section 5.3: It is unclear why the authors only present data on two of the monoterpenes they observed. Did all the monoterpene emissions have large variation between leaves and between trees? They have the data, and a more comprehensive analysis of that data is necessary to demonstrate the capability of this measurement approach for conducting the type of comprehensive, speciated terpenoid analysis that the community needs to better understand factors regulating VOC emission rates from plants. Just showing results for two monoterpenes does not accomplish this.*

We agree with the reviewer that a more thorough example is necessary to highlight the capabilities of this approach. We have chosen to elaborate on the emissions of one of the leaves, so that we can also highlight the atmospheric implications of these measurements. We have added a Figure 10 and its respective caption, below.

We have added this paragraph to Sec. 5.3: "**We provide an example monoterpene emission cassette. Figure 10 puts those emissions into atmospheric context. We show that, although α-pinene contributes to 22% of the measured emissions, it only contributes to 7% of overall OH formation and 0.5% of ozone formation. Although α-terpinene contributes to 20% of the measured emissions, it is the dominating factor in both OH and ozone formation (44% and 98%, respectively). This technique allows for the speciation necessary to understand both the factors which influence emission rates and their subsequent atmospheric impact.**"

We also added this sentence to the last paragraph of Sec. 5.3: "**We further highlight the importance of speciated monoterpene analysis, and this technique's application for such analyses.**"

[Figure]

**Figure 10. Relative measured abundance of all quantified monoterpenes from Ginkgo Leaf 1 (Fig. 9), and the subsequent relative contribution to both OH Reactivity and Ozone Reactivity. OH and ozone reactivity were calculated using $k_{OH}$ and $k_{ozone}$ rate constants from Atkinson (1997).**

*MINOR COMMENTS*

*Regarding emission rate calculation (equation 5, L. 205): how are they estimating leaf surface area? There are a number of different approaches that often vary depending on type of plant. This should be clarified.*

> While there are a number of different approaches for calculating leaf area, all measurements in this manuscript were done with leaves that filled the chamber, resulting in 6 cm$^2$ of leaf area for each measurement. However, we do acknowledge that other applications may not have this convenience.

> We have added the clarifying sentences to the description of equation 5: "**For all measurements in this manuscript, we selected leaves that filled the chamber, for a total measured leaf area of 6 cm$^2$. However, this technique is still applicable for leaves that do not fill the chamber due to size or shape; for such leaves, leaf area mush be determined separately (e.g., via image processing (Chaudhary et al., 2012), or via calculations based on geometric measurements (Sellin, 2000)).**"

*L. 364-365: "though sample collection and analysis is timely." Timely means "opportune" or "prompt". I think the author intends to say something more like time-intensive or labor-intensive.*

> We thank the reviewer for their suggestion and have updated the term "timely" to the intended term "**time-intensive**".

*L. 411: "Temperature response curves can be used". Temperature "response" of what variable? Assimilation? trace gas exchange? All of the above? Unclear what is being referred to.*

> We have clarified that sentence to "Temperature response **of photosynthetic metrics** can be used to compare…"

> We have also added the sentence "**Temperature response of trace gases can be used to further investigate the mechanisms by which different compounds are emitted.**" before "Comparing the emission of lesser-studied compounds like decanal…" in the same paragraph.

*L. 412-413: "For example, this study suggests that basil has a higher photosynthetic thermotolerance than mint despite the fact that basil had a lower CO2 assimilation rate." How so? Can you please elaborate on this referring to specific data in the plots that supports this conclusion?*

> We have expanded upon this statement with the addition of another sentence. "**For example, this study suggests that basil has a photosynthetic maxima at temperatures greater than mint (35 °C, Fig 8; 26 °C, Fig. 7, respectively), despite the fact that basil had a lower overall CO$_2$ assimilation rate. At temperatures above the maxima, photosynthesis and plant productivity may be inhibited (Berry and Bjorkman, 1980), suggesting that basil may have a higher thermotolerance than mint. We note that this comparison only considers short-term temperature increases, and further investigations would be necessary to determine the acclimation potential of these plant species to higher temperatures.**"

*L. 414: "to that of formic acid or monoterpenes can better inform of the impact and deciding factors". 2 comments. There is a typo or grammatical error around "can better inform of the impact". Also, "deciding" doesn't appear to be an appropriate word choice here. Perhaps "regulating"?*

This sentence has been changed to "**Comparing the emission of lesser-studied compounds like decanal to better-studied compounds like formic acid or monoterpenes can improve the understanding of the regulating factors of leaf-level BVOC emissions.**"

*L. 437: "These case study data support that leaf emissions". Typo or grammatical error?*

This terminology has been changed to "**This case study supports previous findings that leaf emissions…**".

*Figure 8: decanal emissions, markers and error bars. If you don't have duplicates to define error bars as standard deviation, I suggest using your analytical error for the error bars. There should be some indication of error, even if it's the small error introduced from the instrumental analysis.*

We thank the reviewer for their observation. We have updated the Figure 8 to reflect instrumental error and have updated the figure caption accordingly.

[Figure]

Figure 8. $CO_2$ assimilation (blue circles) and decanal emission (purple squares) temperature response curve of one basil leaf. $CO_2$ assimilation is fit to a cubic function. We collected $CO_2$ assimilation values ten times over 20 minutes and averaged the values; error bars represent the standard deviation of those measurements. **Error bars for decanal emission represent instrumental error.** The dashed line denotes 0 $\mu g \; m^{-2} \; min^{-1}$ decanal emission.

**Additional References:**

[revised manuscript text omitted]

**Reply to Anonymous Referee #2**

We thank the reviewer for their thoughtful and constructive comments. We have addressed all comments below, and we believe that the manuscript has improved with their suggestions. Reviewer comments are below (grey italicized), with author responses indented. Changes and additions to the manuscript are included, in bold.

*Riches et al. present photosynthesis and trace gas (volatile organic compound, VOC) measurements made using a portable photosynthesis system (PPS) together with chemical ionisation mass spectrometry (CIMS) and offline analysis of sorbent tubes. PPS has been widely used to sample VOCs at well-defined photosynthetic conditions and this manuscript provides a useful characterisation of the LI-COR LI-6800 both in terms of CO2 assimilation and measurement of VOCs. I have a number of minor concerns (listed below), with these points addressed I would recommend publication.*

> We appreciate the reviewer's appreciation of the characterization of this approach, and their involvement with the improvement of this manuscript. We are pleased to address their concerns and comments below.

*SPECIFIC COMMENTS*

*1. Concentration is used throughout when referring to mol mol-1, this should be mole fraction or mixing ratio.*

> We thank the reviewer for pointing this out. Concentration is indeed technically incorrect (though frequently used!); we have replaced all inappropriate uses of this term to 'mixing ratio'.

*2. Different units for emission factor are used when reporting results from the TOF-CIMS and sorbent tubes (moles and mass). It would be easier for the reader if the same units were used throughout.*

> We have converted the formic acid emission rates from nmol $m^{-2}$ $min^{-1}$ to µg $m^{-2}$ $min^{-1}$, and have updated both Figure 7 and in-text discussion of said results accordingly.

> Specifically, paragraph three of Sect. 5.1 was updated to reflect this change: "In contrast, formic acid continues to increase above the photosynthesis maximum, with maximum emission (**2.3 µg $m^{-2}$ $min^{-1}$**)…".

[Figure]

*3. Line 142. Brass fitting are often avoided when working with VOCs, why was a brass hose barb fitting used when all other fittings were stainless steel?*

> The reviewer is certainly correct that the sparing use of inlets is always important! Different VOCs interact differently with different materials; system loss to such materials should always be investigated. Brass fittings are often used sparingly in multiple studies (e.g., Hewitt et al., 2011), and was convenient for our measurements and its use was kept minimal.

*4. Line 147: Is the external pump placed up or downstream of the sorbent tube?*

> Downstream, so that none of the captured VOCs interact with the pump itself. We have added a clarification to the text: "The external pump (Fig. 2E) **is placed downstream of the tube** and ensures constant flow through the sorbent tube."

*5. Line 165: The authors state that the impact of increased flow rates should be investigated for individual species. As a range of species were used in this study, was any effect observed?*

> There was no effect observed for any of the species over the range of flows investigated in this study, however we acknowledge that different volatile organic compounds have vastly different chemical properties, and increased flow rate may effect such chemical species.

*6. Line 191: Most of the TD-GC/MS instrumental methodology is provided in the supplementary information which keeps the manuscript focused on the PPS system. On the whole I appreciate this but would like to see a make and model for the TD-GC/MS in the main text.*

> We agree that the make and model of the TD-GC/MS is important to include in the manuscript and have added the following sentence to the end of Paragraph 1 in Sect. 2.3:
>
> "**Here we use an autosampler (Ultra-xr, Markes Intl.) and thermal desorption unit (Unity-xr, Markes Intl.) coupled to a gas chromatograph (TRACE 1310, Thermo Scientific) mass**

**spectrometer (TSQ 8000 Evo Triple Quadrupole GC-MS/MS, Thermo Scientific).**"

We have also kept those details in the supplementary materials, to maintain thoroughness of Sect. S2.

*7. Line 209: The sampling protocol set out in section 2.4 focuses mainly on sampling to sorbent tubes. How did this differ for the CIMS?*

We thank the reviewer for pointing out this oversight and have added the following sentences to the end of Paragraph 2 of Sect. 2.4:

"**For CIMS measurements, we collect a system blank from the PPS (no leaf) by sampling the SAM subsample port for at least 5 min (at 1 Hz). After enclosing the leaf in the PPS chamber, we monitor the stability of both photosynthesis and volatile emissions. We typically find that the leaf and detector system requires at least 5 min to stabilize.**"

*8. Line 245: Did the authors observe any effect on photosynthesis or VOC emission when keeping a leaf in the chamber for long periods e.g. 8h?*

With leaves in the PPS for extended periods of time, it is still possible to see the effects of the plant's diurnal rhythm. For example, even if a leaf is exposed to constant light and $CO_2$, we observe a dip in photosynthesis towards the evening. Likewise, if samples are collected pre-dawn, even after a leaf has established at a constant light and $CO_2$, we see an increase in both photosynthesis and monoterpene emissions as the rest of the plant is exposed to such conditions. Because the leaf is still attached to the plant, it is still influenced by the conditions of the plant as a whole. Thus, timing of experiments is important to consider, especially when comparing several leaves of a plant using less time-resolved methods like thermal desorption.

What we do not see is a systematic increase or decrease of photosynthesis and emissions as the plant remains in the chamber for an extended period of time. We have added this sentence to Sect. 2.4: "**We observe no consistent evidence that longer periods in the leaf chamber impact photosynthesis or VOC emission.**"

Interestingly, we occasionally (i.e. inconsistently and only on some of the citrus plants) observe, fluctuations in photosynthesis over long periods. These fluctuations vary by up to 3 µmol m$^{-2}$ s$^{-1}$ $CO_2$ assimilation, and can occur at periods on the order of ~10 minutes to ~1 hour. We have excluded data from these periods of time, as they are so inconsistent and we are investigating whether the observations represent plant behavior or are a spurious result from other experimental factors.

*9. Line 268: Does the use of external fans on the PPS impact the rest of the plant?*

We did not observe any effect of the external fans on the rest of the plant. Fans were placed intentionally so as to minimize airflow directed towards the plants, as identified by leaf rustling during a period of otherwise no wind/airflow. To our knowledge, the fan placement had little to no impact on the plants themselves.

*10. Line 310: Is there a temperature dependence in the trace gas background?*

There is a temperature dependence in the trace gas background. For example, formic acid has a temperature dependence in the background. This highlights the importance of regular and

thorough background measurements, especially when establishing temperature and humidity dependence of VOCs, as these parameters effect the entire instrument. We modified the text to read: "The air entering the PPS is ambient, and thus prone to change **throughout the day as sources and sinks vary**."

*11. Line 365: "timely" is a vague term, could you estimate a time?*

Thank you for catching this. As per Reviewer #1's suggestions, we have changed that word to the intended term, "time-intensive".

*12. Line 403: What were the background decanal concentrations? Was this ambient air or was a VOC scrubber used as recommended in section 4?*

These experiments were conducted prior to the introduction of the VOC scrubber, and thus ambient air was used. We have added the following sentence to the end of the first paragraph in Sect. 5.2: "**Background decanal concentrations in ambient air were 11 ± 1 (average ± standard deviation) parts-per-billion.**"

*13. Line 361: The authors describe the observations made in section 5 as a case study rather than emissions ratios to be used in models. While this is true I think that if a more detailed description of the plants used was provided (e.g. growth conditions, develop-mental stage etc.) readers would be able to make more use of the data presented. Perhaps this could be added to the supplementary information.*

We are happy to provide more details about the plants used in the supplementary information. We have such added Sect. S4: "Plant growth conditions" to the supplementary materials.

| Plant | Scientific Name | Developmental Stage | Grow Location | Diel Temperature Variance | Light Conditions |
|---|---|---|---|---|---|
| Basil | *Ocimum basilicum* | Mature, pre-flowering | Indoors, beside north-facing window | ~ 19 – 23 °C | Primarily natural, some fluorescent, standard daylight hours |
| Mint | *Mentha spicata* | Mature, pre-flowering | Indoors, beside north-facing window | ~ 19 – 23 °C | Primarily natural, some fluorescent, standard daylight hours |
| Ponderosa Lemon | *Citrus limon x Citrus medica* | Mature, fruiting | Indoors, greenhouse | ~ 17 – 28 °C | Primarily natural, LED supplement, 16 hours of light |
| Ginkgo | *Ginkgo biloba* | Mature, 2.1 – 2.7 tall | Outdoors, arboretum | ~ 16 – 33 °C | Natural light, standard daylight hours |
| Mulberry | *Morus alba* | Mature, 7.6 – 9.1 m tall | Outdoors, arboretum | ~ 16 – 33 °C | Natural light, standard daylight hours |
| Walnut | *Juglans regia* | Mature, 3.0 – 4.6 m tall | Outdoors, arboretum | ~ 16 – 33 °C | Natural light, standard daylight hours |

*TECHNICAL CORRECTIONS*

*Figure 3: caption refers to "squares, left panel" but the figure shows pentagons*

Figure 3 caption adjusted accordingly.

**Additional References:**

Hewitt, C., Langford, B., Possell, M., Karl, T., and Owen, S.: Quantification of VOC emission rates from the biosphere, TrAC Trends in Analytical Chemistry, 30, 937-944, 2011.

[revised manuscript text omitted]

**S4 Plant growth conditions**

| Plant | Scientific Name | Developmental Stage | Grow Location | Diel Temperature Variance | Light Conditions |
|---|---|---|---|---|---|
| Basil | *Ocimum basilicum* | Mature, pre-flowering | Indoors, beside north-facing window | ~ 19 – 23 °C | Primarily natural, some fluorescent, standard daylight hours |
| Mint | *Mentha spicata* | Mature, pre-flowering | Indoors, beside north-facing window | ~ 19 – 23 °C | Primarily natural, some fluorescent, standard daylight hours |
| Ponderosa Lemon | *Citrus limon x Citrus medica* | Mature, fruiting | Indoors, greenhouse | ~ 17 – 28 °C | Primarily natural, LED supplement, 16 hours of light |
| Ginkgo | *Ginkgo biloba* | Mature, 2.1 – 2.7 tall | Outdoors, arboretum | ~ 16 – 33 °C | Natural light, standard daylight hours |
| Mulberry | *Morus alba* | Mature, 7.6 – 9.1 m tall | Outdoors, arboretum | ~ 16 – 33 °C | Natural light, standard daylight hours |
| Walnut | *Juglans regia* | Mature, 3.0 – 4.6 m tall | Outdoors, arboretum | ~ 16 – 33 °C | Natural light, standard daylight hours |

75 **References**

Brophy, P., and Farmer, D.: A switchable reagent ion high resolution time-of-flight chemical ionization mass spectrometer for real-time measurement of gas phase oxidized species: characterization from the 2013 southern oxidant and aerosol study, Atmos Meas Tech, 8, 2945-2959, 2015.